# Sex as a Prognostic Factor in Systematic Reviews: Challenges and Lessons Learned

**DOI:** 10.3390/jpm11060441

**Published:** 2021-05-21

**Authors:** Elena Stallings, Alba Antequera, Jesús López-Alcalde, Miguel García-Martín, Gerard Urrútia, Javier Zamora

**Affiliations:** 1Clinical Biostatistics Unit, Instituto Ramón y Cajal de Investigación Sanitaria, 28034 Madrid, Spain; cochrane.madrid@ufv.es (J.L.-A.); javierza@gmail.com (J.Z.); 2CIBER Epidemiología y Salud Pública (CIBERESP), 28029 Madrid, Spain; mgar@ugr.es (M.G.-M.); GUrrutia@santpau.cat (G.U.); 3Sant Pau Institute for Biomedical Research (IIB Sant Pau), 08041 Barcelona, Spain; AAntequera@santpau.cat; 4Faculty of Health Sciences, Universidad Francisco de Vitoria, 28223 Madrid, Spain; 5Institute for Complementary and Integrative Medicine, University Hospital Zurich, University of Zurich, CH-8091 Zurich, Switzerland; 6Department of Preventive Medicine and Public Health, University of Granada, 18010 Granada, Spain; 7WHO Collaborating Centre for Global Women’s Health, Institute of Metabolism and Systems Research, University of Birmingham, Birmingham B15 2TT, UK

**Keywords:** sex, gender, prognosis, prognostic factor, systematic review, methods

## Abstract

Sex is a common baseline factor collected in studies that has the potential to be a prognostic factor (PF) in several clinical areas. In recent years, research on sex as a PF has increased; however, this influx of new studies frequently shows conflicting results across the same treatment or disease state. Thus, systematic reviews (SRs) addressing sex as a PF may help us to better understand diseases and further personalize healthcare. We wrote this article to offer insights into the challenges we encountered when conducting SRs on sex as a PF and suggestions on how to overcome these obstacles, regardless of the clinical domain. When carrying out a PF SR with sex as the index factor, it is important to keep in mind the modifications that must be made in various SR stages, such as modifying the PF section of CHARMS-PF, adjusting certain sections of QUIPS and extracting data on the sex and gender terms used throughout the studies. In this paper, we provide an overview of the lessons learned from carrying out our reviews on sex as a PF in different disciplines and now call on researchers, funding agencies and journals to realize the importance of studying sex as a PF.

## 1. Introduction

People are living longer, with one or more health problems; prognosis research is thus vital for explaining and predicting future clinical outcomes in people with existing health conditions. Prognosis research aims to summarize and predict relevant outcomes such as death, recovery, recurrence, disability, or quality of life. In the past 10 years, research on prognosis has rapidly increased [1,2,3,4] along with many novel studies and new methods being developed. However, results from different studies are often contradictory, making it difficult to assess a specific prognostic factor (PF). This is where systematic reviews come into play. Nevertheless systematic reviews of PFs have received little attention by scientists to date. In clinical medicine, we are starting to see a transition from a universal medicine that has a one-size-fits-all approach to personalized medicine. Personalized medicine is a unique individualized approach to treatment based on a patient’s diagnosis and prognosis [5]. This intertwinement has led to theragnostics, which is the connection of diagnosis and therapeutics addressed to people on an individual basis [6]. This novel connection can provide better prognoses relying on specific features, i.e., PFs. Genetic information plays an important role in theragnostics and pharmacogenetics—which is the study of how people respond differently to drug therapies based on their genes—and helps to individually tailor treatments [7]. There are underlying genetic mechanisms to most sex differences in disease [8], which suggests that sex is an excellent candidate as a PF.

Research on PFs is becoming more widespread, and its importance in clinical practice is gradually being recognized. A PF “is any measure that, among people with a given health condition (that is, a start point), is associated with a subsequent clinical outcome (an endpoint)” [9]. Therefore, PFs can distinguish groups of people with a different average prognosis. PF research has a wide variety of applications related to both clinical and public health research [9]. For example, for many cancer patients, tumour grade at the time of diagnosis is a prognostic factor, as each group of patients with the same tumour grade should have broadly similar outcomes [3]. Also, a high body mass index (BMI) is a PF for worse outcomes in patients diagnosed with COVID-19 [10]. Similarly, male sex is a poor PF in non-small-cell lung cancer [11,12] and in gastric cancer, as females experience a better survival rate [13]. On the other hand, female sex is a poor PF for mortality in acute myocardial infarction [14,15]. Thus, minimal clinically relevant differences associated with patients’ sex may have a great impact on the understanding of disease processes, the applicability of the findings to specific patient groups, and the planning of future research [16]. Sex refers to the biological, genetic, and physiological processes that generally distinguish females from males, while gender refers to the roles, relationships, behaviours, and other traits that societies typically attribute to women, men, and people of diverse gender identities (e.g., transgender people) [17]. Sex is, with age, the most common baseline factor collected in the context of randomized and non-randomized studies, regardless of whether a study addresses a therapeutic, etiologic, diagnostic, or prognostic topic. Therefore, sex has clearly the potential to be evaluated as a PF in almost all clinical areas [8,10,11]. This issue is typically assessed in primary studies but is generally not a considered topic in systematic reviews.

Systematic reviews (SRs) are the cornerstone of evidence-based medicine as they play a major role in summarizing the available body of evidence and in identifying knowledge gaps [18]. Accordingly, addressing sex-related findings in systematic reviews is important to better guide clinical practice and tailor patient care to provide the optimal treatment for different sexes. In contrast, in primary studies or systematic reviews, not considering how meaningful such differences between sexes are can lead to poorer healthcare quality, limiting the generalizability of study results and promoting inequities. In recent years, research on sex as a PF has increased [1,2,3,4]; however, this influx of new studies frequently shows conflicting results across the same treatment or disease state.

Sex differs from other PFs in that it is difficult to verify through tests and it generally does not change over time. We have written this article to offer insights into the challenges we encountered when conducting SRs on sex as a PF and to provide suggestions on how to overcome these obstacles, regardless of the outcome or clinical domain. We carried out two systematic reviews on sex as a PF. The first review studied the prognostic role of sex on mortality outcomes in sepsis [19], while the second one looked at the role of sex in the prognosis of patients with acute pulmonary embolism [20]. In the current paper, we will comment on the lessons learned from these two reviews and, in the methodological challenges section, we will present three SRs studying sex as a PF, providing examples and making comparisons. Our objective is to discuss the methodological challenges we encountered and reflect on the lessons learned in carrying out these reviews. Therefore, future systematic reviewers will be able to learn from our experiences and use the same framework whilst investigating sex as a prognostic factor. Primary researchers will also be able to benefit from this paper, as they will understand the difficulties that reviewers encounter when trying to synthesize this type of studies. For example, authors may become aware of what terms should be used in abstracts and titles to maximize the likelihood of their study being captured in review searches.

## 2. Importance of Sex as a PF and Its Demarcation from Gender

Sex is a biological attribute that is associated with physical and physiological features including gene expression, hormone function and reproductive and sexual anatomy [17,21,22]. Sex, typically assigned at birth based on the appearance of external genitalia, is defined as female, male, intersex, etc. However, it is often mislabelled as gender. Sex and gender are interconnected but vastly different. In comparison to sex, gender refers to the socially constructed roles, behaviours and identities of female, male and gender-diverse people [22] and to the terms men and women or boys and girls. Both sex and gender play roles as prognostic factors in various illnesses (cardiovascular disease, sepsis, cancer) [15,19,23]. Therefore, it is important to distinguish between them when studying sex or gender as a prognostic factor.

Many differences exist between the sexes, mostly due to genetics and hormones (different levels of androgens and oestrogens). Many illnesses are characterized by a higher incidence in one sex versus the other. For example, 99% of people diagnosed with breast cancer are females [24]. In the same manner, four times more females than males are diagnosed with osteoporosis [25]. Other illnesses will occur at the same rate in both sexes, but they can manifest differently according to sex. For example, in schizophrenia, the disease usually starts at an earlier age and with severer symptoms in males [26]. In myocardial infarctions, again the first myocardial infarction is experienced at a younger age by males than by females, and females tend to present with symptoms of nausea and shortness of breath instead of the usual chest pain. These sex differences in disease incidence and in diagnosing illness also predict differences in prognosis. Just as differences have been found in the manifestation of myocardial infarction in females, it has also been found that females tend to have poorer outcomes [27,28,29]. Similarly, in our review, we found an independent prognostic impact of sex on mortality, although in this case the certainty of evidence was very low [19].

Prognosis research has increased in the past decade, and the same can be said of sex and gender research [1,2,3,4]. However, in general, there are not many SRs on PFs. For example, when we searched the Cochrane database of systematic reviews, we found three completed PF SRs [30,31,32]. In comparison to intervention reviews, which is the more traditional review type with thousands of reviews completed, this is a novel area of research and evidence synthesis. Thus, taking into consideration that so few reviews have been published on PFs and sex separately and that both areas of research are rapidly developing, it is understandable that there is a lack of SRs on sex as a PF. However, we did find a few SRs evaluating the role of sex, such as Bougouin et al., Giuliano et al. and Kim et al., in various illnesses, and thus we were able to compare and contrast these reviews and review the methods that they used [33,34,35].

## 3. Methodological Challenges

### 3.1. Search and Selection

A search of reviews on sex as a PF retrieves thousands of references; therefore, it is important to use a search filter. We added a sex and gender filter (“sex factors” OR “sex distribution” OR “Sex characteristics” OR “Sex ratio” OR sex OR “women’s health” OR “men’s health”) OR TITLE: (boy* OR male* OR girl* OR female* OR gender OR women OR men OR sex) to the search strategy in an attempt to narrow down the search field. By adding the filter, the number of studies retrieved from the searches was reduced by 20%. In combination with searching electronic databases, we also hand-searched conferences. For conferences on sex and gender, we found the congress “Organization for the study of sex differences”. We hand-screened 8 years of abstracts from this conference and did not retrieve any study that met all inclusion criteria. While retrieving unpublished studies from conference abstracts is considered good practice in systematic review development, it is important to consider the expected large number of results from the electronic database searches. Thus, we encourage review authors to choose to extend their search to conference proceedings based on their resource availability.

When screening studies, both sexes must be included for a study to be eligible for inclusion, as it is impossible to measure the prognostic significance of sex while only looking at one sex. As mentioned above, sex refers to genotypic, phenotypic and physiological characteristics, including chromosomes, gene expression, hormone levels and reproductive and/or sexual anatomy [17]. In our reviews, we accepted any assessment of sex and evaluated the appropriate use of the terms sex and gender when applicable. We were aware that the terms ´sex´ and ´gender´ are poorly described and defined in the majority of published articles. Thus, when no additional information was provided, if it was clear that the authors were referring to sex but mistakenly used the terms for gender, we assumed that the study was considering sex. If the authors explicitly stated that they evaluated the social aspect, then we considered that they were evaluating gender and not sex.

### 3.2. Data Extraction 

For data extraction, we used the CHARMS-PF (critical appraisal and data extraction for systematic reviews of prediction modelling studies for prognostic factors) template [9]. CHARMS-PF is a checklist of key data to be extracted from primary PF studies. It is based on additions and modifications of the original CHARMS data extraction sheet for prediction modelling [36]. In the CHARMS-PF extraction sheet, there is a section created for PFs (index and comparator factors). In this section, we extracted the PF definition and method of measurement of the PF. We accepted any definition of sex (our PF of interest) and any method of sex measurement given by the authors. The timing of PF measurement does not matter when studying sex in primary studies or reviews, as it is not normally a temporal variable that may change. We extracted information on the use of the terms sex and gender in each study to evaluate if the terms were being used adequately in the primary studies. These data are important to extract and take note of, as the lack of literacy surrounding the terms for sex and gender should be highlighted in SRs.

Bougouin et al. did not use CHARMS for data extraction [33]. However, CHARMS, is relatively new (2014) and was only published a year before the publication of this review [36]. Thus, the authors may had previously planned their data extraction methods in a protocol and did not change them. However, they did extract the adjusted data, though they did not extract many data on gender, their prognostic factor of interest. Giuliano et al. did not mention the use of CHARMS but created their own data extraction template [35]. Kim et al. also did not use CHARMS, as it was not yet published [34].

### 3.3. Risk of Bias Assessment

A critical step in carrying out a systematic review is assessing the risk of bias of the included studies. Tools used to measure quality are ROB and ROB2 for randomized trials and PROBAST for prediction model studies [37,38,39,40]. To assess the risk of bias in PF studies, the “Quality in Prognosis Studies” (QUIPS) tool was created [41,42]. The tool consists of several prompting questions within six different domains, each domain being judged on a three-grade scale. Hayden et al. determined six key domains for the risk of bias appraisal included in PF studies: study participation, study attrition, PF measurement, other prognostic factor adjustment, outcome measurement and analysis and reporting [42].

We used an amendment to the QUIPS tool proposed by Aldin and colleagues [30] using four categories (low, moderate, high and unclear risk) instead of the initial three categories (low, moderate and high). In SRs of sex as a PF, the unclear category may be especially relevant, since some signalling items of QUIPS, such as those related to PF domains with a high likelihood of lack of sex definition, have a limited value for the assessment and rating. Therefore, rating as unclear risk may be the fairest alternative. Following on from the rating amendment, we also made some slight modifications to the QUIPS sections to adapt it for sex as a PF, which are highlighted in Table 1. Some items were particularly hard to differentiate, and a learning phase was required to increase the interrater agreement.

In contrast, Bougouin et al. did not use QUIPS for quality assessment, but again this could be due to the short time frame between QUIPS and the review being published [33]. Giuliano et al. used QUIPS to assess the risk of bias of the included studies; however, they did not mention any modifications being made to the tool [35]. Kim et al. did not use QUIPS in their assessment of risk of bias, as it was not yet published [34].

### 3.4. Data Analysis

The studies incorporated in a systematic review of sex as a PF need to be similar in terms of population (ages, ethnicity, etc.), index factor measurement (sex) and outcome measurement (how the outcomes are measured, for example, mortality in 30 days, 90 days, etc.): a meta-analysis must combine the results from sufficiently homogenous individual studies to provide a meaningful pooled prognostic effect estimate. If the studies are not homogenous in design, we may carry out a subgroup analysis or meta-regression. Studies may report the crude association between PF, sex, and the outcome or the adjusted association, where one adjusts for the contribution of other PFs compared to the index factor (here, sex). However, we did not require the consideration of the complete core set of additional PFs. In our experience, if the researchers adjusted for at least one of our pre-defined confounding factors, then the study was valid for inclusion in the review and the meta-analysis. Deciding the pre-defined confounding factors was complicated, as sex is a factor that is present from birth; therefore, it is complex to define what is a confounding factor of sex. To make a list of the most important confounding factors, we created a Delphi panel of reviewers and clinicians to decide on which factors to include. Our data extraction and risk of bias assessments considered the confounding factors that were measured, controlled for (by the study design) and adjusted for (in the analysis).

An additional part of the analysis in reviews studying sex as a PF is to analyse the sex/gender terminology used. To judge if the terms sex and gender were used correctly in the primary studies, we conducted a frequency analysis of the results. In our SR on sex as PF in patients with sepsis, we included 13 studies [19]. No primary study included in our review defined sex correctly. Twelve of the included studies in our review had an inadequate use of sex and gender terms, using all the terms interchangeably throughout the study. The correct usage of terms was unclear in the remaining study, as it used the term gender and all the related terminology for gender; however, from the study context, it could be presumed that the study authors were in fact referring to sex [43].

In some SRs, there are discrepancies and interchangeability of the sex and gender terms. The correct terms for sex are male and female, and those for gender are boy, girl, man or woman. Many published reviews use the word gender when referring to sex, thus making it confusing for readers. Authors feel that they are being repetitive and do not realise that sex and gender are two distinct terms. For example, in Bougouin et al. and Kim et al., the authors use the terms gender, men and women consistently but in reality, they are discussing topics related to sex, not gender [33,34]. In Kim et al., the authors state “women tend to have smaller coronary arteries than men” [34]. This is a sex difference, not a gender difference and should instead read “female patients tend to have… than male...”. Likewise, Giuliano et al. talk about sex differences whilst referring to men and women. In other parts of the paper, they also use the terms male and female, thus making their usage of the terminology incorrect and inconsistent [35].

## 4. Concluding Remarks

The role of sex in human health and medical research continues to be understudied, as sex-based medicine is often viewed as a specialist niche instead of being central to all medical research [16]. We must bring sex- and gender-based analysis to the forefront of research and base future research around it. Systematic reviews evaluating the role of sex as a PF fosters rigorous, reproducible, inclusive and responsible science.

When carrying out a PF SR with sex as the index factor, it is important to keep in mind the adaptations that must be made in various SR stages. This is outlined in Table 2 below and includes modifying the PF section of CHARMS-PF, adjusting certain sections of QUIPS and extracting data on the terms sex and gender used throughout the studies. The lack of literacy regarding the sex and gender terms needs to be addressed, as this is a widespread problem among researchers. It is especially important that researchers wishing to study and publish sex and gender research understand the differences between these concepts and use the correct terminology. This lack of understanding can have serious implications in prognosis research, such as creating confusion among investigators and the general public.

There are methods available to rigorously synthesize the role of sex as a PF. We hope to see more systematic reviews of this kind in the future. In this paper, we have provided an overview of the lessons we learned from carrying out our reviews on sex as a PF in different disciplines and we now call on researchers, funding agencies, journals and research institutions to acknowledge the importance of studying sex as a PF. Realizing this importance is a critical step in the right direction towards precision medicine that will help reduce health inequities and benefit both males and females alike.

## Figures and Tables

**Table 1 jpm-11-00441-t001:** QUIPS modifications for studying sex as a prognostic factor.

Domains	QUIPS	QUIPS Modified for Sex as PF	Comments
1. Study participation	Description of the baseline study sample	Baseline number and characteristics of participants by sex are clearly described and reported separately for males and females	The regular QUIPS refers to a description of the baseline sample in general (both sexes combined); however, we specified that it was necessary to have the participants characteristics described by sex. Example: Females (N): race of females (N), obesity in females (N). Males (N): race of males (N), obesity in males (N).
2. Study attrition	Adequate description of participants lost to follow-up	Key characteristics of participants lost to follow-up are provided separately for males and females	The key characteristics of the lost-to-follow-up participants must be recorded by sex. N of females and N of males per characteristic. However, this was never reported.
3. Prognostic factor measurement	Clear definition or description of the PF	Clear definition or description of sex	The authors must provide an adequate definition for the prognostic factor, in this case sex ^1^.
	Adequately valid and reliable method of measurement	Not applicable	We do not anticipate specific sex measurement for this type of research question.
	Continuous variables reported or appropriate cut points used	Not applicable	Sex measurement is not a continuous variable.
	Same method and setting of measurement used in all study participants	Not applicable	We do not anticipate method and setting measurement for this type of research question.
	Adequate proportion of the study sample had complete data	Not applicable	We do not anticipate missing data of sex measurement for this type of research question.
	Appropriate methods of imputation were used for missing data	Not applicable	We do not anticipate missing data of sex measurement for this type of research question.
4. Outcome measurement		No differences in this domain.	
5. Adjustment for other prognostic factors		No differences in this domain.	
6. Statistical analysis and reporting		No differences in this domain.	

^1^ We considered an adequate definition as listing any of the following: sex for biological characteristics; gender for socially constructed roles, behaviours, and identities; females or males for sex; women or men for gender.

**Table 2 jpm-11-00441-t002:** Summary of challenges and solutions in a systematic review evaluating the role of sex as a prognostic factor.

	Challenge	Solution
1. Search and selection	Too many references retrieved in the search.	Add a sex and gender search filter to the search
2. Data extraction	Sections of CHARMS-PF not totally compatible with sex as a prognostic factor (PF)	Take the following into consideration: Accept any definition of sex and any method of sex measurement given by authors Timing of PF measurement is not important, as sex is not normally a temporal variable that may change Extract data on the use of the terms sex and gender
3. Risk of Bias	Sections of QUIPS not compatible with sex as a PF	Specific modifications in QUIPS tool as listed in Table 1
4. Data analysis	Deciding the confounding factors for sex as a PF	Delphi panel (expert input) to aid in this decision-making process
	Sex and gender terms used inadequately and interchangeably in many primary studies	Analyse the sex and gender terminology used in primary studies

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
