# Peer review of "Sex as a Prognostic Factor in Systematic Reviews: Challenges and Lessons Learned"

_jpm, 2021, doi:10.3390/jpm11060441_

Round 1

Reviewer 1 Report

The paper ”Sex as a prognostic factor in systematic reviews: Challenges and lessons learned” by Stallings and colleagues presents methodological considerations based on two former systematic reviews of this research group (refs. #40,41), exemplified by three recent publications (refs. #34-36). The message is clear, but the role of the two groups of publications (#34-36 vs. #40,41) should be sharpened. To this end, the last paragraph of the introduction (i.e. l.95-98 and l.132-133) should state that the lessons were learned from earlier own endeavors, whereas the three recent examples are used for the sake of exemplification. At least, this is how I read their respective roles.

The manuscript type is, therefore, a mixture of “Communication” or “Methodological aspects” and a narrative review. I propose to change the article type in line 1 to either “Communication” or “Methodology” for the sake of clarity, as this is not an original research endeavor (though practical experiences are, of course, original). “Commentary” or “Opinion” will also do.

L.108-9: I fear that many authors use “sex” and “gender” interchangeably for the sake of variation of the language. One possibility to further strengthen this point would be to enhance the title of Section 2, for instance “2. Importance of sex as PF and its demarcation from gender” (l.99).

The references are comprehensive and represent an interesting and illuminating collection on the topic.

Minor corrections:

L.6-19, affiliations: instead of listing each author’s affiliations singularly, all affiliations should be listed according to their chronological appearance, starting with those of the first author:

1Clinical Biostatistics Unit, Instituto Ramón y Cajal de Investigación Sanitaria, Madrid, Spain.

2CIBER Epidemiología y Salud Pública (CIBERESP), Spain.

Right now, CIBERESP happens to appear 5 times.

L.45,50,170,175,203: delete superfluous hyphens: reviews instead of re-views; ad-dressed; do-mains; inter-rater; re-view

L.49: a patient’s diagnosis

L.85: I guess “how meaningful are such differences between the sexes” should read “how meaningful such differences between the sexes are,”

L.87-88: “this influx of new studies frequently shows”

L.91: delete the comma (“to give insights … and to provide”)

L.100: “Sex is the biological attribute that is associated”

L.102: delete “is” and the following becomes, indeed, an inserted sentence; “Sex, typically assigned…., is characterized…”

L.117: “experiencing the first heart attack” instead of “for the first heart attack”

L.124,130: delete apostrophes – PFs, SRs instead of PF’s, SR’s

L.131: instead of publication years, use “et al.”, here: “…, such as Bougouin et al., Giuliano et al. and Kim et al. [34-36] in various illnesses, ….” and delete “[34-36]” in l.133.

L.132: illnesses,  

L.150,154,211,213,216: “et al.” instead of years of publication, thanks.

l.150-155, 178-181: there is no notion of ref. #35. Please add.

L.159-160: refs. #38,39 refer to PROBAST. Please add accordingly those for ROB and ROB2.

L.184-186: different font type in footnote (sans serif) in opposition to the rest of the manuscript (Times New Roman).

L.194: “In our experience, ”

L.199: delete the comma

L.210: “do not” instead of the colloquial “don’t”, thanks.

Reviewer 2 Report

Thank you for giving me an opportunity to review this paper. In this study, authors have given an a picture of the lessons learned from conducting their reviews on sex as a PF in different disciplines. Moreover, they are trying to get researchers, funding agencies, and journals attention to realize the importance of studying sex as a PF. However, the way the presented the information is bad. After reading this paper several times I could not find main point. Methodology and challenges part was worse. They tried to discussed other people work not their work, what they have done. I did not find any paragraph reading challenges. I don't know what they are trying to say by providing table 1. There is no studying findings, directly jumped to conclusion remark which was not well organized.

Reviewer 3 Report

This review brings an interesting topic related to sex as a prognostic factor when conducting a systematic review. The review will help the reader understand more about this important factor. However, in my opinion, many parts are needed to improve the manuscript quality as well as provide more specific information. Please refer to the following comments:

  • I would suggest the study content should be more comprehensive regarding challenges in the whole step of systematic review, not just data extraction and quality assessment. For example, from the searching query or establishing the question, inclusion and exclusion criteria for this specific indicator? In addition, which databases or library that we should commonly search for these type of topic. Each database has a specific topic and limited coverage ability. How the author should handle these problems to overcome the missing-study problem?
  • Since the manuscript focus on a study that uses sex as a prognostic factor. In my opinion, evidence such as searching these studies, giving an overview on general pitfall would be helpful for future study. Although the study aims to provide the difficulties and solutions when conducting a systematic review of the study having sex as a prognostic factor. There was no information of evidence or references saying that these challenges is common.
  • Regarding risk of bias assessment, sex as a prognostic factor is not only included in the clinical trial or prediction model study but also commonly conducted with the observational study. Therefore, a comprehensive review in terms of the quality assessment tool is better than just mention a few quality assessment tools.
  • About the data analysis, the study mentioned the incorrect way of using sex or gender. In my opinion, it would be nice if there is a table about it such as a table with common mistakes and solutions.
  • In general, the topic is of interest to me, however, the manuscript content needs to be more specific to increase the scientific soundness as well as the quality. For example, in data analysis section, the manuscript was written “a meta-analysis must combine the results from sufficiently homogenous individual studies to provide a meaningful pooled prognostic effect estimate”, I wonder what is the homogenous criteria in this sentence? Any criteria to evaluate it when conducting a meta-analysis and how homogenous is enough to conduct a meta-analysis with the study having sex as a prognostic factor? Besides, the study is not always homogenous in terms of result or study design, so what kind of heterogeneity is usually appears, any solution to deal with it or should we exclude all? Excluding all heterogeneity without reasonable criteria will potentially lead to the skew result or misleading also.
  • In addition, in line 195, what is the meaning of pre-defined confounding factors here? Any criteria for these pre-defined factors, if there is more than one factor, how do we define which factor is more important?

In general, the topic of this review is interesting however, more comprehensive work should be done to improve the quality as well as contribute to the improvement of further study.

Round 2

Reviewer 2 Report

Thanks for the revised manuscript. It is still unclear and confusing. The authors did not provide any information on how many previous articles they have evaluated and which challenges they have faced. They mentioned that "Primary researchers will also be able to benefit from this paper as they can understand the difficulties that reviewers encounter when trying to synthesize these types of studies.”--What kinds of benefit will get from your study findings?. what is the clinical implications of this research?.

Reviewer 3 Report

The manuscript content has been improved from the first version. However, in my opinion, the study content can be improved more as I expect it should have more specific information regarding sex as prognostic factor.
In addition, I would refer additional example such as example from author previous work. 
Next, since this is a lesson learn paper, a summarization table would be nice to provide the important notes as well as solution describing through out of the manuscript. 
